# Vaccination in Aged Care in Australia: A Retrospective Study of Influenza, Herpes Zoster, and Pneumococcal Vaccination

**DOI:** 10.3390/vaccines13070766

**Published:** 2025-07-20

**Authors:** Stephen Wiblin, Yuen Lai, Natalie Soulsby, Jodie Hillen

**Affiliations:** 1Pfizer Australia Pty Ltd., Sydney, NSW 2000, Australia; yuen.lai@pfizer.com; 2Embedded Health Solutions, Melbourne, VIC 3194, Australia; natalie.soulsby@embeddedhealth.com.au; 3Quality Use of Medicines and Pharmacy Research Centre, University of South Australia, Adelaide, SA 5000, Australia; jodie.hillen@unisa.edu.au

**Keywords:** influenza vaccines, pneumococcal vaccines, herpes zoster vaccines, vaccine-preventable diseases, homes for older people

## Abstract

**Background**: Older adults living in aged care are at risk of poor health outcomes due to influenza, pneumococcal disease, and herpes zoster infections. Despite these conditions being vaccine-preventable, little is known about vaccine uptake rates in the residential elderly care setting in Australia. **Methods**: This was a retrospective cohort study examining the medical records of residents of 31 aged care homes in Australia (n = 1108). Data were extracted from medical records for the period March 2023 to September 2023. The proportion of residents vaccinated against influenza, pneumococcal disease, and herpes zoster was calculated. Univariate and multivariate logistic regressions were used to identify possible demographic and other characteristics associated with the vaccination uptake. **Results**: This study included 1108 residents. Two-thirds (68%) were female, and the median age was 87 years. All residents had one or more comorbidities. Most (92.6%) had received an influenza vaccine within the prior two years, but only 38.3% had received a pneumococcal vaccine, and 16.8% had received herpes zoster vaccination. In all models, receipt of the other vaccines was a significant predictor for vaccine administration. The other factor associated with influenza vaccination was non-consumption of alcohol and younger age for herpes zoster vaccination. **Conclusions**: While there is a high uptake of influenza vaccines, there is a low uptake of both pneumococcal and herpes zoster vaccines in residents of aged care facilities. Further research into the barriers and enablers of vaccine uptake should be undertaken, with the goal of increasing the vaccination uptake in this vulnerable population.

## 1. Introduction

Older adults living in residential elderly care homes experience a multitude of risk factors, like comorbidities, concomitant disabilities, and age-related immunosenescence [1]. These factors make them more vulnerable to infections and at increased risk of severe infectious disease, which can result in hospitalisation and death [1]. Outbreaks of vaccine-preventable diseases in elderly care homes can have major health and economic consequences [2]. The incidence of herpes zoster increases with age, owing to a progressive decline in virus-specific cell-mediated immunity. In Australia, between 2006 and 2013, there were an estimated 13.7, 15.3, and 19.9 cases per 1000 population per year in the 60–69 years, 70–79 years, and ≥80 years age groups, respectively. Older people are also more likely to develop complications due to zoster virus reactivation [3]. Reducing the burden of vaccine-preventable diseases in Australian elderly care homes provides an opportunity to promote positive ageing, prevent suffering, and improve quality of life, and it is consistent with the Immunisation Agenda 2030 that ‘all people benefit from recommended immunizations throughout the life course, effectively integrated with other essential health services’ [4].

In Australia, the National Immunisation Programme (NIP) recommends influenza vaccination for all adults annually; pneumococcal vaccination is funded for people aged 70 years and over and for those with additional medical risk conditions; and herpes zoster vaccination is funded for people aged 65 years and over and for people with additional medical risk conditions (given as a single dose prior to 1 November 2023) [5,6]. These provide protection both to the individual and the community (including other elderly care residents) from these diseases. Vaccines on the NIP are provided by the Australian Government for free to people who are eligible for Medicare. However, there are significant barriers to the vaccination of older adults, which include access to care, mobility, multiple providers, a lack of provider confidence in adult vaccination, and a culture of paediatric-only immunisation [2]. An additional barrier may be the location of older people’s care homes relative to a general medical practice. This impacts the time to deliver care and has previously been highlighted as a barrier to receiving general practice services in elderly care [7].

The Australian Immunisation Register (AIR) is the national register that records all vaccines administered to all people living in Australia under the NIP, through school programmes or privately (such as for influenza or travel) [8]. For adults, influenza vaccinations since March 2021, and other NIP vaccinations have been recorded since July 2021 [9]. The data so far show that the uptake of influenza vaccine in Australians aged 65 years or older is 46%, and, as an example, the uptake of zoster vaccine in the 70 to 79-year age group is 31% [10]. However, there are currently no reliable data on the uptake of pneumococcal vaccination in older Australians due to the complexity of the dosage regimens [10]. Further, vaccination figures presented are likely to substantially underestimate true uptake, due to the identification of major gaps between the number of doses distributed and those recorded on AIR [10].

Even less is known about the vaccination rates of people living in elderly care homes. A small study conducted in four elderly care homes in Sydney reported an influenza vaccination rate of 84%; however, this study was not representative of the Australian elderly care population, primarily due to the small sample size [11]. There is no publicly available information regarding the uptake of the other recommended vaccines in Australian elderly care homes.

The purpose of this study was to determine vaccination rates for influenza, pneumococcal, and herpes zoster vaccinations and to determine if there are any explanations for the vaccination uptake for people living in residential elderly care facilities. Improved explanations for the acceptance of vaccines could help inform the design of a pharmacist-led vaccine programme in aged care.

This article is a revised and expanded version of a paper which was presented at the Communicable Diseases & Immunisation Conference 2025 in Adelaide, South Australia [12].

## 2. Materials and Methods

### 2.1. Study Design and Setting

This was a retrospective cohort study examining the medical records of residents of 31 elderly care facilities in Australia. Data were extracted from the medical records from March 2023 to September 2023 (sample selection window), with a look-back period to 1 January 2017. The study was conducted in accordance with legal and regulatory requirements, as well as with scientific purpose, value, and rigour, and followed generally accepted research practices described in the Guidelines for Good Pharmacoepidemiology Practices (GPP) issued by the International Society for Pharmacoepidemiology (ISPE). The study was overseen by internal ethics from Embedded Health Solutions.

### 2.2. Participants

To be included in the study, participants needed to reside in an Australian-registered elderly care facility, be eligible (by age) to receive Australian Government-funded vaccinations, and have previously provided written informed consent to access historical medical records for the purpose of the quality review of medicines. Resident records were excluded if the resident was deceased prior to the start of data extraction. They were not excluded if they died during the sampling period.

### 2.3. Variables

Any influenza, pneumococcal disease, or herpes zoster vaccination that was documented in the resident’s medical notes during the study period was recorded, regardless of the date of administration of that vaccination. This is important for those vaccinations that are given as ‘once in a lifetime’ or with two doses administered in close succession, without the requirement for booster vaccination.

The data collection timeframe represented a reasonable historical review and coincides with when Australia transitioned from the Australian Childhood Immunisation Register (ACIR) to the AIR, although the mandatory recording of vaccines on the NIP was not introduced until 2021. Pneumococcal vaccines (PPV23, PCV13, or both) and herpes zoster vaccines given at any time were recorded. Only influenza vaccines given within the last 2 years were recorded.

In addition to the vaccination status, participant demographics were collected from the resident’s medical record for use in univariate and multivariate modelling. Clinically relevant risk factors or comorbidities that were associated with worse outcomes were chosen. These included age, race, sex, socioeconomic status of the facility location (Socioeconomic Indexes for Areas [SEIFA] decile), geographical region, recent hospitalisation (within 30 days), length of stay at aged care home, comorbidities, smoking and alcohol consumption (drinks/day), and use of oral corticosteroid in the last 3 months [11,13,14]. Chronic steroid use was chosen as a confounding variable, as it predisposes residents to viral or bacterial infection due to its immunosuppressive effects. Information on confounders was collected from administrative data available at the residential care home.

All data collected were deidentified. Residential care homes were chosen and intended to represent a maximally diverse, wide demographic sample encompassing socioeconomic, ethnic, religious, and other demographic considerations.

The study was not designed to collect safety data. Any adverse events that were detected by the credentialed pharmacist who was assigned to the aged care home on review of medical records were reported as per local practice.

### 2.4. Statistical Methods

This study planned to include up to 1500 residents. To reduce bias, random sampling of elderly care facilities from 350 elderly care homes serviced by Embedded Health Solutions occurred during routine clinical pharmacist medication review visits. The interrater reliability was managed through the validation of data by a second clinical pharmacist. Homes were stratified by SEIFA, and then randomly selected using a random allocator, ensuring that homes within each SEIFA category were represented.

Data are summarised descriptively. A univariate logistic regression model was built to examine the influence of SEIFA decile, age at data collection, comorbidity burden, sex, other vaccination status, alcohol consumption, smoking, recent hospitalisation, and corticosteroid use on the uptake of vaccination. Covariates that were significant at *p* < 0.20 were entered into a multivariate logistic regression model (Section 3.3).

Analyses were conducted using R version 4.0.1 (2020-06-06) and Stata MP v18 for Mac (StataCorp., College Station, TX, USA).

This study has been reported according to RECORD-PE [15].

## 3. Results

### 3.1. Participants

There were 1108 residents from 31 residential elderly care facilities included in the study. Facilities were located in New South Wales (n = 10), Victoria (n = 7), Queensland (n = 7), and South Australia (n = 7). Two-thirds (68%) of included residents were female, and the median (range) age was 87 years (71 to 107 years) (Table 1 and Appendix A, Figure A1).

Residents in NSW were considered the most disadvantaged, and those in Victoria were considered the most advantaged (Table 1). Just over 1 in 10 had used corticosteroids in the last 3 months, but rates were higher in Victoria, at 19.2%. Approximately 8% had been hospitalised in the last 30 days (Table 1).

All residents had one or more comorbidities. The most commonly reported comorbidities were mental and behavioural disorders (927/1108, 83.7%), diseases of the circulatory system (924/1108, 83.4%), and diseases of the musculoskeletal and connective tissue (802/1108, 72.6%). Respiratory conditions, including asthma, chronic obstructive pulmonary disease, or bronchiectasis, affected 262 residents (23.6%). The median number of comorbidities was 10 (range 3, 30).

### 3.2. Vaccination Rates

Overall, 1026/1108 (92.6%) elderly care residents had received an influenza vaccine within the prior two years (Table 2 and Figure 1). However, the rate of uptake of other recommended vaccinations was much lower: only 424 (38.3%) had ever recorded a pneumococcal vaccine, and only 184 residents (16.8%) had ever recorded herpes zoster vaccination. Only 11.5% of the residents had received all three vaccines, with 31.5% receiving two vaccines, and 50.1% receiving one of the vaccines. Just under 7% had received no vaccinations at all. There was a difference in vaccine administration rates by state, with zoster vaccination being less likely in Victoria and pneumococcal vaccination being less likely in NSW or Victoria compared to other states (Table 2). While there was no difference in the vaccination rates by the number of reported comorbidities for the influenza vaccine and zoster vaccine, those with higher numbers of comorbidities were more likely to receive vaccination (HR 1.03, 95% CI 1.00 to 1.06, *p* = 0.04).

### 3.3. Univariate

Significant variables in the univariate model (at *p* < 0.20) for influenza vaccination included SEIFA decile, pneumococcal vaccination status, zoster vaccination status, alcohol use, recent hospitalisation, and corticosteroid use (Table 3). There was no difference in vaccination rates by sex. Specifically, for every 1 unit increase in the SEIFA decile, the odds of receiving an influenza vaccine were 9% lower (OR 0.91; CI 0.84 to 0.98). If a resident had received a pneumococcal vaccine, they were more than ten times more likely to have also received an influenza vaccine (OR 10.63; 95% CI 4.27 to 26.49). Similarly, if a resident had received a herpes zoster vaccine, they were nearly nine times more likely to have also received an influenza vaccine (OR 8.77; 95% CI 2.14 to 36.01).

**Table 3 vaccines-13-00766-t003:** Univariate and multivariate modelling.

Model		Univariate			Multivariate	
Covariate	Influenza n = 1108	Pneumococcal n = 1108	Herpes zoster n = 1108	Influenza n = 990 *	Pneumococcal n = 1094 *	Herpes zoster n = 990 *
Socioeconomic Indexes for Areas (SEIFA) decile	OR 0.91 [95%CI 0.84, 0.98] *p* = 0.013	OR 1.00 [95%CI 0.96 1.04] *p* = 0.846	OR 0.94 [95%CI 0.89, 1.00] *p* = 0.038	OR 0.93 [95%CI 0.85, 1.00] *p* = 0.062	-	OR 0.95 [95%CI 0.89, 1.01] *p* = 0.095
Age (years)	OR 1.00 [95%CI 0.97, 1.03] *p* = 0.955	OR 0.98 [95%CI 0.97, 1.00] *p* = 0.052	OR 0.88 [95%CI 0.86, 0.91] *p* < 0.001	-	OR 1.01 [95%CI 0.99, 1.03] *p* = 0.348	OR 0.88 [95%CI 0.85, 0.90] *p* < 0.001
Gender	OR 1.02 [95%CI 0.63, 1.65] *p* = 0.930	OR 1.11 [95%CI 0.85, 1.43] *p* = 0.447	OR 0.77 [95%CI 0.56, 1.07] *p* = 0.123	-	-	-
Influenza vaccination	-	OR 10.63 [95%CI 4.27, 26.49] *p* < 0.001	OR 8.77 [95%CI 2.14, 36.01] *p* = 0.003	-	OR 8.60 [95%CI 3.43, 21.56] *p* < 0.001	OR 10.19 [95%CI 1.36, 76.32] *p* = 0.024
Pneumococcal vaccination	OR 10.63 [95%CI 4.27, 26.49] *p* < 0.001	-	OR 5.01 [95%CI 3.55, 7.08] *p* < 0.001	OR 14.35 [95%CI 4.46, 46.13] *p* < 0.001	-	OR 4.44 [95%CI 3.01, 6.55] *p* < 0.001
Herpes zoster vaccination	OR 8.77 [95%CI 2.14, 36.01] *p* = 0.003	OR 5.01 [95%CI 3.55, 7.08] *p* < 0.001	-	OR 8.73 [95%CI 1.18, 64.36] *p* = 0.033	OR 4.85 [95%CI 3.36, 7.01] *p* < 0.001	-
Alcohol use	OR 0.58 [95%CI 0.32, 1.03] *p* = 0.076	OR 1.18 [95%CI 0.81, 1.70] *p* = 0.387	OR 1.65 [95%CI 1.06, 2.57] *p* = 0.026	OR 0.51 [95%CI 0.27, 0.97] *p* = 0.039	-	OR 1.62 [95%CI 0.97, 2.69] *p* = 0.065
Hospitalisation in the last 30 days	OR 0.61 [95%CI 0.30, 1.23] *p* = 0.165	OR 0.79 [95%CI 0.50, 1.25] *p* = 0.316	OR 0.47 [95%CI 0.22, 0.98] *p* = 0.044	OR 0.64 [95%CI 0.29, 1.38] *p* = 0.253	-	OR 0.53 [95%CI 0.24, 1.19] *p* = 0.122
Corticosteroid use in the last 3 months	OR 1.79 [95%CI 0.76, 4.19] *p* = 0.181	OR 1.12 [95%CI 0.77, 1.62] *p* = 0.555	OR 0.99 [95%CI 0.61, 1.61] *p* = 0.960	OR 1.59 [95%CI 0.66, 3.84] *p* = 0.306	-	-

Note: intercepts not presented. * may not equal the full 1108 residents, where there were missing data on included covariates.

Significant variables in the univariate logistic regression models for pneumococcal vaccination uptake (at *p* < 0.20) were age, influenza vaccination status, and zoster vaccination status (Table 3). There was no difference in vaccination rates by sex. If a resident had been vaccinated for influenza, they were more than ten times more likely to have also received a pneumococcal vaccine (OR 10.63; 95% CI 4.27, 26.49), and if a resident had received a herpes zoster vaccine, they were five times more likely to have also received a pneumococcal vaccine (OR 5.01; 95% CI 3.55, 7.08).

Significant variables in the univariate logistic regression model for herpes zoster vaccination uptake (at *p* < 0.20) were SEIFA decile, age, sex, influenza vaccination status, pneumococcal vaccination status, alcohol use, and hospitalisation in the last month (Table 3). The strongest predictors of herpes zoster vaccination were younger age and having received a pneumococcal vaccine.

Corticosteroid use did not seem to influence pneumococcal or zoster vaccination rates.

### 3.4. Multivariate

In the multivariate logistic regression model for the influenza vaccine uptake, those with a pneumococcal vaccination were 14.4 times more likely, and those with herpes zoster vaccination were 8.7 times more likely, to receive influenza vaccination than those who had not received these other vaccinations. Residents who consume alcohol were 49% less likely to receive an influenza vaccination (Table 3).

In the multivariate logistic regression model for the pneumococcal vaccine uptake, those who had received an influenza vaccination or herpes zoster vaccination were 8.6 times and 4.9 times more likely to have also received the pneumococcal vaccination (Table 3).

In the multivariate logistic regression model for herpes zoster vaccination, for every yearly increase in age, residents were 12% less likely to have received a herpes zoster vaccination. However, residents who had received an influenza vaccination or pneumococcal vaccination were 10.2 times or 4.4 times more likely to have also received the herpes zoster vaccination, respectively (Table 3).

## 4. Discussion

Of the 1108 residents from a broad cross-section of Australian elderly care facilities reviewed, 1026 (92.6%) had received influenza vaccination in the past two years. Only 424 (38.3%) had ever received a pneumococcal vaccine, and only 184 residents (16.8%) had ever received herpes zoster vaccination. Our findings are supported by Victorian data from 2022, which showed that the median uptakes for influenza, pneumococcal, and herpes zoster vaccines were 86.8%, 32.8%, and 19.3%, respectively, [16].

In all the multivariate models in our study, the receipt of another vaccination was a significant predictor of vaccination. Our findings might suggest that vaccine use is behavioural and receiving one vaccine reflects willingness to receive multiple vaccines. Alternatively, it may be related to the aged care home’s or GP’s motivations, behaviours, or practices and policies. As such, low rates of vaccination for pneumococcal disease or herpes zoster, despite a willingness to receive the influenza vaccine, highlight an opportunity for programmes to increase awareness of the availability and benefits of these recommended vaccines under the NIP. Accredited Clinical Pharmacists are well placed to lead such a programme in Australian elderly care, as they are credentialled to immunise. The Australian Government’s ‘My Health Record’ now includes a direct link to the Australian Immunisation Register to obtain the immunisation status of an individual, which facilitates further audits of vaccination status.

Interestingly, the socioeconomic decile was not significant in any of the multivariate regression models, despite its association in the univariate model influenza and herpes zoster vaccination, with more disadvantaged and more remote residents more likely to receive vaccination. The reasons for this are unclear, but there may be interactions between the SEIFA decile and other explored and unexplored covariates. Elsewhere, a higher socioeconomic status is positively associated with vaccine uptake in the elderly [17]. In other regions of the world, a greater influenza vaccine uptake has been reported in female residents and those who did not smoke, but a significant variability in uptake by facility has also been reported [18]. Other reasons cited in the literature for the poor uptake of recommended vaccinations in the general population include poor health literacy, a lack of understanding of the criteria for receiving immunisations, general practitioners (GPs) not having a reminder system in place for eligible patients, low awareness of personal susceptibility, perceived side effects, and lack of awareness of a residents’ vaccination status [19,20,21].

The apparent willingness of elderly care residents to be vaccinated is evidenced by the high uptake rates for the influenza vaccine. However, there needs to be a campaign to raise awareness of the benefits of being vaccinated against pneumococcal disease and herpes zoster for the older population. In the general population, over the 3-year period from 2014 to 2016, there were 10,613 hospitalisations and 337 deaths related to influenza, 2219 hospitalisations and 20 deaths for invasive pneumococcal disease, and, for shingles, an average of 2473 hospitalisations and 28 deaths per year [22]. Given this burden, it is important that the barriers and facilitators to vaccine administration are urgently explored and addressed, particularly how to operationalise pneumococcal and herpes zoster vaccinations in this setting. In addition, better surveillance of vaccine-preventable infections, such as the policy implemented in Victoria [23], may improve awareness and highlight the importance of vaccination in this vulnerable population.

There are solutions. Firstly, vaccination of the elderly does not need to be viewed with the same lens as childhood vaccination: there are important differences in immune system functioning between children and the elderly, as there are with regard to vaccine efficacy [24]. Once this paradigm shift has occurred, it opens avenues for implementing novel programmes that increase the vaccine uptake. For example, GP software programmes could include the capacity to identify eligible patients for these and other vaccines. It may also be prudent to trial an intervention where residents and their families are informed of the benefit of recommended vaccinations and then offered on-site vaccination, perhaps via a community pharmacy which supplies medications to aged care facilities or as part of a wider resident and employee vaccination programme. Employee vaccination programmes must also be considered to protect vulnerable residents in elderly care, especially given that outbreaks are usually linked with the pathogen being introduced to the facility from employees or visitors [25]. This is also important given the low influenza vaccination uptake by elderly care workers (especially in the pre-COVID-19 era) [26]. Pharmacist-led employee vaccination programmes have been successful in improving the vaccination rates for elderly care health workers [27], although whether such programmes serve to prevent vaccine-preventable disease in residents is still a matter of debate [28].

A strength of this study was the validity of data sources, with medical record abstraction considered the gold standard in pharmacoepidemiology studies [29]. The methodology used does not suffer from some possible limitations, such as linked electronic data or dispensing data. Our study has several limitations. Data were collected from several sources. Possible explanations for missing records of vaccination include a lack of documentation in the handover letter to the aged care home, residents being unaware of their vaccination status, and information not being recorded on the AIR before it was made a mandatory requirement in 2021. The presumption was made that if it was not recorded in any of those sources, then the vaccine had not been given. Therefore, rates of vaccination may be underestimated, particularly for vaccines that are given infrequently (for example, pneumococcal and herpes zoster vaccines). This study included residents of 31 elderly care facilities that received clinical pharmacy services from Embedded Health Solutions. Their vaccine-related policies are written by the facility providers and may have differed from those of other Australian elderly care providers, which could potentially limit the generalisability of the results. The analysis was limited by the sample size. Further, data collection for this study occurred during the COVID-19 pandemic, when awareness of and uptake of the influenza vaccine was high [30]. In 2020, influenza vaccine administration was recorded in the AIR as 64.0% for adults aged ≥ 65 years, 6.0 percentage points higher than in 2019, and true coverage is likely to be higher [30]. Vaccine uptake was being strongly encouraged for all elderly care residents, and reporting of influenza vaccination is mandatory. This may partially explain the very high proportion of elderly care residents vaccinated against influenza. It could be interesting to observe the subsequent effect of influenza vaccination reverting to a voluntary choice and any impact of post-COVID-19-pandemic vaccine fatigue in this population.

## 5. Conclusions

A comprehensive review of 1108 residents across Australian elderly care facilities revealed significant disparities in the vaccination rates for recommended immunizations. While the influenza vaccination showed an excellent uptake, at 92.6% over two years, the pneumococcal vaccination rates were concerningly low, at only 38.3%, and herpes zoster vaccination was even lower (16.8%).

This study identified a strong behavioural pattern, where residents who received one vaccination were significantly more likely to receive others, suggesting either an individual willingness to be vaccinated or facility-level practices that promote comprehensive immunisation programmes. These findings highlight missed opportunities to improve protection against vaccine-preventable diseases in this vulnerable population.

Despite the burden of disease—with pneumococcal disease causing over 2000 hospitalizations annually and herpes zoster resulting in approximately 2500 hospitalizations and 28 deaths per year—the uptake remains suboptimal. The high acceptance of influenza vaccination demonstrates that elderly care residents are generally willing to be immunised, indicating that barriers to pneumococcal and herpes zoster vaccination are likely system-related rather than due to vaccine hesitancy.

The research suggests several practical solutions, including enhanced awareness campaigns, GP software reminders for eligible patients, on-site vaccination programmes through community pharmacies, and comprehensive resident and staff vaccination initiatives. Accredited Clinical Pharmacists are particularly well-positioned to lead such programmes given their immunisation credentials and regular presence in elderly care facilities.

Addressing these vaccination gaps represents a critical opportunity to reduce preventable hospitalizations and deaths in elderly care populations through systematic implementation of evidence-based immunisation strategies.

## Figures and Tables

**Figure 1 vaccines-13-00766-f001:**
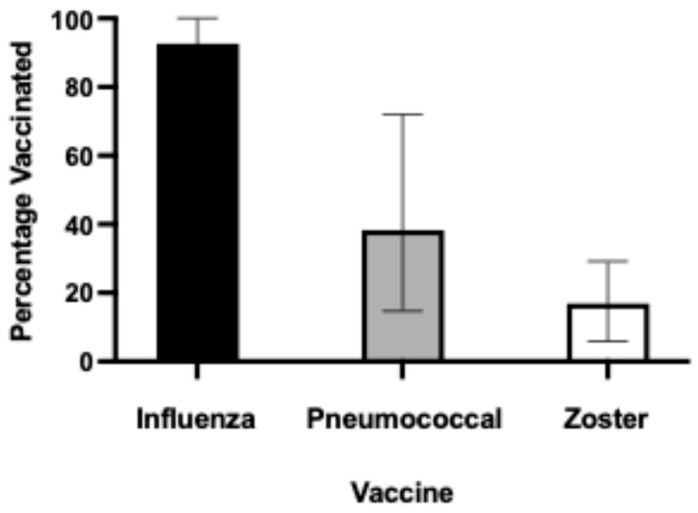
Percentage of elderly care residents vaccinated against influenza, pneumococcal disease, or herpes zoster.

**Table 1 vaccines-13-00766-t001:** Demographics and characteristics of residents.

	Level	NSW	Victoria	Queensland	SA	Total
N		347	250	264	247	1108
Age at data collection (years), mean (SD)		86.2 (6.9)	86.4 (6.9)	87.3 (7.0)	86.9 (7.3)	86.7 (7.0)
Age at data collection (years)	70–74	22 (6.4%)	19 (7.6%)	6 (2.3%)	14 (5.7%)	61 (5.5%)
	75–79	46 (13.3%)	26 (10.4%)	40 (15.2%)	36 (14.7%)	148 (13.4%)
	80–84	63 (18.2%)	46 (18.4%)	40 (15.2%)	44 (18.0%)	193 (17.5%)
	85–89	100 (28.9%)	71 (28.4%)	64 (24.2%)	46 (18.8%)	281 (25.4%)
	90–94	78 (22.5%)	58 (23.2%)	75 (28.4%)	74 (30.2%)	285 (25.8%)
	95+	37 (10.7%)	30 (12.0%)	39 (14.8%)	31 (12.6%)	137 (12.4%)
Gender	Female	228 (65.7%)	175 (70.0%)	173 (65.5%)	172 (69.6%)	748 (67.5%)
	Male	119 (34.3%)	75 (30.0%)	91 (34.5%)	75 (30.4%)	360 (32.5%)
SEIFA Decile, median (range)		4.0 (1.0, 10.0)	7.0 (2.0, 10.0)	6.0 (1.0, 9.0)	5.0 (2.0, 9.0)	5.0 (1.0, 10.0)
Corticosteroid use in the last 3 months		43 (12.4%)	48 (19.2%)	29 (11.0%)	13 (5.3%)	133 (12.0%)
Hospitalisation in the last 30 days		34 (9.8%)	20 (8.0%)	24 (9.1%)	12 (4.9%)	90 (8.1%)

SEIFA = Socioeconomic Indexes for Areas.

**Table 2 vaccines-13-00766-t002:** Summary of vaccinated residents.

	NSW	Victoria	Queensland	SA	Total
N	347	250	264	247	1108
Influenza vaccine given	319 (91.9%)	235 (94.0%)	235 (89.0%)	237 (96.0%)	1026 (92.6%)
Herpes zoster vaccine given	61 (18.1%)	23 (9.2%)	49 (18.8%)	51 (20.6%)	184 (16.8%)
Pneumococcal vaccine given	111 (32.0%)	70 (28.0%)	140 (53.0%)	103 (41.7%)	424 (38.3%)

## Data Availability

The data that support the findings of this study are available on request from the corresponding author, SW, following appropriate ethical review and approval. The data are not publicly available as participants of this study did not give written consent for their data to be shared publicly.

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
