# Peer review of "Vaccination in Aged Care in Australia: A Retrospective Study of Influenza, Herpes Zoster, and Pneumococcal Vaccination"

_vaccines, 2025, doi:10.3390/vaccines13070766_

Round 1
Reviewer 1 Report
Comments and Suggestions for Authors
Wiblin et al., present a retrospective study focusing on vaccination rates in Australian aged care settings. They emphasize on the importance of vaccine uptake in the vulnerable population. The authors observed high uptake of the flu vaccine as opposed to both pneumococcal and herpes zoster shots in aged care residents. The article is well described and the paragraph highlighting possible solutions and study limitations is valuable. The manuscript would benefit if the following points were addressed:
1. The study only looked at residents of "Embedded Health Solutions". Generalizing the findings to the whole country could be a possible stretch, as certain vaccine-related policies applied might be unique to this particular facility. Adding this to the limitations or future directions could be an option.
2. Fig 1 can be broken into age bins and geographical sites as images are more useful in conveying such data compared to a tabular format
3. Have the vaccination rates changed post covid, has the data been collected/analyzed? If not, are there any plans for a future study? This could be touched upon as well.
Author Response
|
Comment 1: Wiblin et al., present a retrospective study focusing on vaccination rates in Australian aged care settings. They emphasize on the importance of vaccine uptake in the vulnerable population. The authors observed high uptake of the flu vaccine as opposed to both pneumococcal and herpes zoster shots in aged care residents. The article is well described and the paragraph highlighting possible solutions and study limitations is valuable. The manuscript would benefit if the following points were addressed: 1. The study only looked at residents of "Embedded Health Solutions". Generalizing the findings to the whole country could be a possible stretch, as certain vaccine-related policies applied might be unique to this particular facility. Adding this to the limitations or future directions could be an option. |
|
Response 1: Thank you for pointing this out. We have added text to the limitations section of the study. Per below. Please also note, line 129-131 described that the study was intended to be maximally diverse and not wholly generalizable. Also, Embedded Health Solutions (EHS) are a company that provides clinical pharmacy services into aged care homes and EHS do not have vaccine related policies as they are written by the homes or providers. The data was from a cross section of 31 aged care homes serviced by EHS. Added text at lines 303-305: ‘The study included residents of 31 aged care facilities that received clinical pharmacy services by Embedded Health Solutions. Their vaccine related policies are written by the facility providers and may have differed to those of other Australian aged care providers which could potentially limit the generalisability of the results.’ |
|
Comment 2: Fig 1 can be broken into age bins and geographical sites as images are more useful in conveying such data compared to a tabular format |
|
Response 2: Thank you for the suggestion, however little additional value will be gained from representing the more granular data graphically. |
|
Comment 3 Have the vaccination rates changed post covid, has the data been collected/analyzed? If not, are there any plans for a future study? This could be touched upon as well. |
|
Response 3: Thank you for the comment. Data were collected in 2023, with a look back to 2017. While data has been collected for date of vaccine, only the most recent influenza vaccine information was collected. So, it’s not possible to see if vaccination rates have changed post COVID with current data but could be something to do in future. To address this, a note about future studies has been added into the discussion. Added text at line 314-316: ‘It could be interesting to observe the subsequent effect of influenza vaccination reverting to a voluntary choice, and any impact of post-COVID-19-pandemic vaccine fatigue in this population.’ |
Reviewer 2 Report
Comments and Suggestions for Authors
Sample
“To be included in the study, participants needed to reside in an Australian registered aged care facility; and be eligible (by age) to receive Australian Government funded vaccinations; and have provided written informed consent.’
[It is extremely important to understand how you selected your participant homes. Please provide your selection method and the % they constitute of the total number of care home residents in Australia]
Discussion
I have 3 questions:
- You summarise factors in the literature that may relate to vaccination status. Do you have the data for your sample?
- Can you provide the infection rates for the care facilities you studied for the illnesses for which you provide the vaccination status?
- Nursing homes are part of a larger medical network in which care facility patients are referred out to hospitals for urgent or semi urgent care then received back. Do you have the infection rates in the hospitals in the states that the nursing homes are in?
You provide no explanations of the rates except by inferences from the literature.
“In all the multivariate models in our study, receipt of another vaccination was a significant predictor for vaccination. Our analysis might suggest that vaccine use is behavioural and receiving one vaccine reflects willingness to receive multiple vaccines. Alternatively, it may be related to the aged care home’s or GP’s motivation, behaviours, or practices and policies. As such, low rates of vaccination for pneumococcal disease or herpes zoster, despite willingness to receive influenza vaccine highlights an opportunity for programs to increase awareness of the availability and benefits of these recommended vaccines under the NIP. Accredited Clinical Pharmacists are well placed to lead such a program in Australian aged care, as they are credentialled to immunise. The Australian Government ‘My Health Record’ now includes a direct link to the Australian Immunisation Register to obtain the immunisation status of an individual, which facilitates further audits of vaccination status.”
“Elsewhere, higher socioeconomic status is positively associated with vaccine uptake in the elderly [16]. In other regions of the world greater influenza vaccine uptake has been reported in female residents, and those who did not smoke; but also reported significant variability in uptake by facility [17]. Other reasons cited in the literature for poor uptake of recommended vaccinations in the general population include poor health literacy, a lack of understanding of the criteria for receiving immunisations, general practitioners (GPs) not having a reminder system in place for eligible patients, low awareness of personal susceptibility, perceived side effects, and lack of awareness of a residents’ vaccination status [18-20].”
Author Response
|
Comments 1: Sample |
|
Response 1: Thank you for pointing this out. We disagree with this comment. Therefore, we have provided the following explanation: The intended study population is designed to be maximally diverse and included a selection of residential care homes from a wide range of geographical and diverse areas. This was outlined in the Methods in Section 2.3 Variables lines 129-131. Articulating the number of sites as a % does not add any additional value to the study manuscript. |
|
Comments 2: You summarise factors in the literature that may relate to vaccination status. Do you have the data for your sample? |
|
Response 2: The study didn’t seek to specifically capture these data, moreover this was a reflection of the previous literature. The study captured a range of demographic details and conclusions where appropriate or supported by study data and these have been made already within the manuscript. Further, understanding attitudes to vaccine uptake in our population was not an objective of this study. |
|
Comments 3: Can you provide the infection rates for the care facilities you studied for the illnesses for which you provide the vaccination status? Nursing homes are part of a larger medical network in which care facility patients are referred out to hospitals for urgent or semi urgent care then received back. Do you have the infection rates in the hospitals in the states that the nursing homes are in? You provide no explanations of the rates except by inferences from the literature. “In all the multivariate models in our study, receipt of another vaccination was a significant predictor for vaccination. Our analysis might suggest that vaccine use is behavioural and receiving one vaccine reflects willingness to receive multiple vaccines. Alternatively, it may be related to the aged care home’s or GP’s motivation, behaviours, or practices and policies. As such, low rates of vaccination for pneumococcal disease or herpes zoster, despite willingness to receive influenza vaccine highlights an opportunity for programs to increase awareness of the availability and benefits of these recommended vaccines under the NIP. Accredited Clinical Pharmacists are well placed to lead such a program in Australian aged care, as they are credentialled to immunise. The Australian Government ‘My Health Record’ now includes a direct link to the Australian Immunisation Register to obtain the immunisation status of an individual, which facilitates further audits of vaccination status.” “Elsewhere, higher socioeconomic status is positively associated with vaccine uptake in the elderly [16]. In other regions of the world greater influenza vaccine uptake has been reported in female residents, and those who did not smoke; but also reported significant variability in uptake by facility [17]. Other reasons cited in the literature for poor uptake of recommended vaccinations in the general population include poor health literacy, a lack of understanding of the criteria for receiving immunisations, general practitioners (GPs) not having a reminder system in place for eligible patients, low awareness of personal susceptibility, perceived side effects, and lack of awareness of a residents’ vaccination status [18-20].” |
|
Response 3: Thank you for raising this. However, the study was not powered to report on rates of infection, rather it was capturing rates of vaccinated patients. Infection rates for associated hospitals were not collected, nor an objective of this study. To meet MDPI requirements for publication, rates of infection can be included from national surveillance databases if absolutely necessary. |
Reviewer 3 Report
Comments and Suggestions for Authors
I have finished the review of the manuscript "Vaccination in aged care in Australia: a retrospective study of influenza, herpes zoster and pneumococcal vaccination", it is an interesting retrospective study indicating association between influenza, pneumococcal and zoster vaccine application in a two-year period in aged, institutionalized, patients with healthcare coverage.
Even when interesting, there aspects that must be further clarified,
Please provide ethical statement, which Board of ethics reviewed and approved? please provide how did you obtained consent from patients, given that more than 90% have mental disabilities? were caregivers or legal tutors informed? Old people, cared at institutions are considered vulnerable population by international bioethics guidelines and requiere protection.
Most of the introduction leads on the importance of influenza and pneumococcal vaccination, but only twice are zoster vaccination mentioned, in lines 45 and 79, indicating it is funded and that uptake can be provided in the 70-79 age group. Adding that authors mentioned asking about a two year period, How can they present as a result if the patients had ever had zoster vaccination? this is an important mislead, if the recommendation is for a specific age group, patients are unable to recall and authors only reviewed last two-years, then it should not be stated the way it is now, because it is confusing and invites the reader to doubt the results and the methods leading to them.
The aim in lines 73-75 is also confusing, as authors did not design a predictive model, but an explanatory one by establishing associations retrospectively and obtaining Odds Ratios.
If only one should Zoster be applied between 70-70 YO and no annual vaccination recommendation exists, isn´t any analysis on zoster vaccination biased? at least, the probability of complete coverage for pneumococcal or influenza doubles the one for zoster, because it is not supposed to be annual, is it?
Lines 93, 94 do you mean eliminated? what happened with records who died during data extraction? unless authors could preview who was going to die and who was not.
Lines between 93 and 106 represent a deviation for the previously explained methods, it seems that authors relied in any notes mentioning zoster vaccine application? it is not an acceptable source, lacks rigor and it was systematically stated in records as mentioned, please remember that absence of evidence does not mean evidence of absence.
Additionally, if originally it was programmed to look back to 2017, why were authors reviewing historical records of the patients? is this admissible? please mention which IRB approved these methods. When permission is provided to search records within a period, at least to my knowledge, only by amendment approved by the IRB can other periods or information sources be consulted.
Line 125 mentions that 1,500 patients were to be included, please provide the sample size calculation and the data entered to obtain the 1500 sample size; also, please provide a reason for including less than the projected sample and mention this limitation in the correspondent section. Please indicate how not establishing a minimum sample size would affect the inference based on these results.
Please indicate how did you asses interrater variability in record reviewing, how many record reviewers obtained the information and how you treated with any missing data.
Figure 1 indicates "proportion" in the title, but presentes percentage in the axis.
Tables must be improved, cells have too much information, layout is confusing, font is too big and there is too much blank space. Please consider using a professional service for table formatting.
Discussion includes zoster only marginally and when addressing influenza vaccination, authors should consider is zoster vaccination in this paper and with the used methods is relevant, and is they decide to include it, they should truly contrast and contextualize zoster vaccination as a measure in a two year period in patients aged 87 in average, when the recommendation mentioned in the introduction is to have an uptake between 70-79 years.
Authors´ contributions (lines 299-304) must follow CrediT format and Vaccine journal guidelines.
Line 311, mentions that participants provided informed consent, so, did they? even when having mental disabilities and being vulnerable population? Please explain.
Author Response
|
Comments 1: Please provide ethical statement, which Board of ethics reviewed and approved? please provide how did you obtained consent from patients, given that more than 90% have mental disabilities? were caregivers or legal tutors informed? Old people, cared at institutions are considered vulnerable population by international bioethics guidelines and require protection |
|
Response 1: Thank you for pointing this out. We disagree with this comment. Therefore, we have provided the following explanation: Embedded Health Solutions (EHS) have an internal ethics review board that oversaw the ethical governance of this study. All data collected from patients was done so with existing consent in place to access medical records, and in this case vaccination status. Informed consent was obtained from all study participants, and legal guardians where required. This consent is pre-existing to access medical records for the purpose of quality review of medicines. The study did not involve human subjects, other than review of records and there is not a requirement for any additional ethics review. To our knowledge, according to the NHMRC, people residing in aged care homes are not considered vulnerable however they are considered to have a power imbalance. This study did not target people with cognitive impairment nor assess this in any way. The study did not collect data directly from residents. The study collected medical record data as per quality improvement protocols at aged care facilities which is covered by internal governance and ethics. |
|
Comments 2: Most of the introduction leads on the importance of influenza and pneumococcal vaccination, but only twice are zoster vaccination mentioned, in lines 45 and 79, indicating it is funded and that uptake can be provided in the 70-79 age group. Adding that authors mentioned asking about a two year period, How can they present as a result if the patients had ever had zoster vaccination? this is an important mislead, if the recommendation is for a specific age group, patients are unable to recall and authors only reviewed last two-years, then it should not be stated the way it is now, because it is confusing and invites the reader to doubt the results and the methods leading to them. |
|
Response 2: We agree with parts of this comment and have, accordingly, modified the introduction also include and emphasize the importance of Zoster vaccination. Added text at lines 41-44 ‘Herpes zoster incidence increases with age owing to a progressive decline in virus-specific cell-mediated immunity. In Australia, between 2006 and 2013, there were an estimated 13.7, 15.3, and 19.9 cases per 1,000 population per year in the 60–69 years, 70–79 years, and ≥80 years age groups, respectively. Older people are also more likely to develop complications due to zoster virus reactivation.’ However, this study used medical records which is the gold standard for medical information and therefore is not at risk of recall bias. We have amended the results text to read ‘ever recorded’ rather than ‘ever received’ in line 172-173 in order to resolve this comment. We have also noted in the discussion lines 293-296 that a strength of the study is the methodology of manual medical record abstraction. |
|
Comment 3 The aim in lines 73-75 is also confusing, as authors did not design a predictive model, but an explanatory one by establishing associations retrospectively and obtaining Odds Ratios. |
|
Response 3: Thank you for this comment. This has been amended in the manuscript line 82. |
|
Comment 4: If only one should Zoster be applied between 70-70 YO and no annual vaccination recommendation exists, isn´t any analysis on zoster vaccination biased? at least, the probability of complete coverage for pneumococcal or influenza doubles the one for zoster, because it is not supposed to be annual, is it? |
|
Response 4 The study sought to determine any recorded vaccination, including for the zoster vaccine. All records were searched dating back to 2017, and any results for the zoster vaccine were included in the manuscript. The authors are unaware of any bias this proposes. |
|
Comment 5 Lines 93, 94 do you mean eliminated? what happened with records who died during data extraction? unless authors could preview who was going to die and who was not. |
|
Response 5 Records were included if the resident died during the data extraction sampling period, however any records of deceased residents were excluded if the death had occurred prior to the start of data extraction. We have made a minor edit to Methods 2.2 Participants lines 103-105 to clarify this point. |
|
Comment 6: Lines between 93 and 106 represent a deviation for the previously explained methods, it seems that authors relied in any notes mentioning zoster vaccine application? it is not an acceptable source, lacks rigor and it was systematically stated in records as mentioned, please remember that absence of evidence does not mean evidence of absence. |
|
Response 6: we agree with this statement and have discussed the strengths and limitations of the sources in lines 292-301. We acknowledge that the Australian immunization register (AIR) was an incomplete source of data, as it was not mandatory to record vaccines into the register until 2021. So, for the periods before this, the study researchers had to rely on where this information may have otherwise been recorded. Review of medical notes was another confirmatory source of data. |
|
Comment 7: Additionally, if originally it was programmed to look back to 2017, why were authors reviewing historical records of the patients? is this admissible? please mention which IRB approved these methods. When permission is provided to search records within a period, at least to my knowledge, only by amendment approved by the IRB can other periods or information sources be consulted. |
|
Response 7. We offer the following explanation. The study was based on retrospective review of medical records back to 2017. Embedded Health Solutions deemed the study a quality improvement program not requiring additional ethical approval. |
|
Comment 8: Line 125 mentions that 1,500 patients were to be included, please provide the sample size calculation and the data entered to obtain the 1500 sample size; also, please provide a reason for including less than the projected sample and mention this limitation in the correspondent section. Please indicate how not establishing a minimum sample size would affect the inference based on these results. |
|
Response 8: Although the sample size is below the initial intended sample, the included residents still represent a maximally diverse sample across a range of demographic areas. 1108 were included of a planned 1500. There was not a formal sample size calculation. We have added in the discussion, line 307 that ‘the analysis was limited by the sample size' |
|
Comment 9: Please indicate how did you asses interrater variability in record reviewing, how many record reviewers obtained the information and how you treated with any missing data. |
|
Response 9: Interrater reliability was managed through embedded health solutions. The clinical pharmacist who collected the data, had the data validated by a second clinical pharmacist supervisor. Text to this effect has been added to Methods Section 2.4, lines 138-139. |
|
Comment 10: Figure 1 indicates "proportion" in the title, but presents percentage in the axis. |
|
Response 10. Thank you for highlighting this. It has been amended in the figure legend. |
|
Comment 12 Tables must be improved, cells have too much information, layout is confusing, font is too big and there is too much blank space. Please consider using a professional service for table formatting. |
|
Response 12 Amended and changed layout and font size in the manuscript. |
|
Comment 13 Discussion includes zoster only marginally and when addressing influenza vaccination, authors should consider is zoster vaccination in this paper and with the used methods is relevant, and is they decide to include it, they should truly contrast and contextualize zoster vaccination as a measure in a two year period in patients aged 87 in average, when the recommendation mentioned in the introduction is to have an uptake between 70-79 years. |
|
Response 13 The reference in the introduction was the uptake in this age range at the time of reporting (line 65). That is not to say that residents aged greater than 79 cannot have received the vaccine. It is not unreasonable for this to be included in the manuscript. However to address the mention of zoster it has been amended to be more explicitly stated in the introduction line 69. Please also see Response 2 for comment on including zoster vaccine in the study. |
|
Comment 14 Authors´ contributions (lines 299-304) must follow CrediT format and Vaccine journal guidelines |
|
Response 14. Format of author contributions statement has been amended in lines 347-351 |
|
Comment 15: Line 311, mentions that participants provided informed consent, so, did they? even when having mental disabilities and being vulnerable population? Please explain. |
|
Response 15 We have made a minor edit to Methods 2.2 Participants lines 103-4 to clarify this point. Please also refer to Response 1 where we explain that consent to access medical records, in this case vaccination status was already in place. This consent was obtained from all study participants, and their legal guardians where required. |
|
4. Response to Comments on the Quality of English Language N/A |
|
Additional response. We would also like to take this opportunity to note that manual medical record abstraction is the gold standard in pharmacoepidemiology studies but not often used due to resource constraints and this is why linked data sets etc and electronic medical records are commonly used to increase sample sizes. However, this comes at a cost of more inferences, for example dispensing data does not confirm actual administration of a medicine etc.
|
Reviewer 4 Report
Comments and Suggestions for Authors
The article entitled “Vaccination in aged care in Australia: a retrospective study of influenza, herpes zoster and pneumococcal vaccination” is a retrospective study to determine vaccination rates for influenza, pneumococcal, and herpes zoster vaccinations and to determine if there are any predictors of vaccination uptake. This study has many limitations and needs to consider the following points:
- Introduction: The authors referred to the purpose of their study; however, they didn’t explain the significance of it. Add a short paragraph to summarize this point
- Discussion: Authors need to explain the following points in their discussion:
- Data collected during COVID-19 (2023), when influenza vaccination was heavily promoted,may inflate influenza uptake compared to pre-pandemic trends. Justify this point
- Discuss COVID-19’s impact on vaccine attitudes (e.g., heightened awareness vs. fatigue)
- The conclusion is too short. Add more details to summarize the main points and findings, emphasizing the research's significance.
- References are written in the wrong format. Correct them to follow the MDPI format
- Linguistic mistakes:
Some words are overused, like vaccine uptake. Use other words like administration.
Replace "abstinence from alcohol" with "non-consumption of alcohol."
Our analysis might suggest = Our findings might suggest
In the captions of table 1, define the medical terms like SEIFA.
Author Response
|
Comments 1: Introduction: The authors referred to the purpose of their study; however, they didn’t explain the significance of it. Add a short paragraph to summarize this point |
|
Response 1: Thank you for pointing this out. We agree with this comment and have added a sentence to that effect to the introduction. Added text at line 83-85: ‘Improved explanations for acceptance of vaccines could help inform the design of a pharmacist-led vaccine program in aged care.’ |
|
Comments 2: Discussion: Authors need to explain the following points in their discussion:
|
|
Response 2: Thank you for these suggestions for the discussion. In the submitted manuscript the point about promotion of influenza vaccine during COVID-19 at lines 307-309 was worded: ‘Further, data collection for this study occurred during the COVID-19 pandemic, when awareness of and uptake of the influenza vaccine was high’ and has been referenced to Beard et al 2021 [1],. The discussion continued (lines 311-313): ‘Vaccine uptake was being strongly encouraged for all aged care residents, and reporting of influenza vaccination is mandatory. This may partially explain the very high proportion of aged care residents vaccinated against influenza’. To further explain and justify – yes influenza vaccination was being promoted through public messaging during the period captured by the study. As demonstrated by Beard et al rates of uptake were higher than in 2019. They say: ‘Vaccine uptake was recorded as 64.0% for adults aged ≥ 65 years, 6.0 percentage points higher than in 2019. While true coverage in older adults is likely 10 to 20 percentage points higher, based on knowledge of vaccine doses distributed and previous national data,4–6 completeness of reporting to AIR has steadily improved with coverage increasing from 31.5% in 2017 and 46.3% in 2018.’ We have inserted a sentence to this effect at lines 309-311. In terms of the impact of COVID-19 on vaccine attitudes, we feel it may be premature to comment, as at the time the data was collected, the impact on attitudes was unknown. We do acknowledge that in aged care it was mandatory to have the influenza vaccine early in the pandemic, but now there is more vaccine fatigue being seen and the uptake is much lower. This is reflected in the addition of text lines 314-316: ‘It could be interesting to observe the subsequent effect of influenza vaccination reverting to a voluntary choice, and any impact of post-pandemic vaccine fatigue in this population.’ |
|
Comment 3: The conclusion is too short. Add more details to summarize the main points and findings, emphasizing the research's significance. |
|
Response 3: Thank you for the opportunity to extend the concluding statements. We have considered some additional items, such as the impact of COVID-19 here, but would reiterate that although the study has limitations, it continues to highlight the importance of up to date vaccination in older and more vulnerable populations, and that this study confirms that vaccine coverage in this population remains suboptimal, and future studies addressing the role of pharmacist led vaccination services should be considered. Here is the revised expanded conclusion, per lines 318-341: ‘A comprehensive review of 1,108 residents across Australian aged care facilities revealed significant disparities in vaccination rates for recommended immunizations. While influenza vaccination showed excellent uptake at 92.6% over two years, pneumococcal vaccination rates were concerningly low at only 38.3%, and herpes zoster vaccination was even lower at 16.8%. The study identified a strong behavioral pattern where residents who received one vaccination were significantly more likely to receive others, suggesting either individual willingness to be vaccinated or facility-level practices that promote comprehensive immunization programs. These finding highlights missed opportunities to improve protection against vaccine-preventable diseases in this population vulnerable to such infections. Despite the burden of disease - with pneumococcal disease causing over 2,000 hospitalizations annually and herpes zoster resulting in approximately 2,500 hospitalizations and 28 deaths per year - uptake remains suboptimal. The high acceptance of influenza vaccination demonstrates that aged care residents are generally willing to be immunized, indicating that barriers to pneumococcal and herpes zoster vaccination are likely system-related rather than due to vaccine hesitancy. The research suggests several practical solutions, including enhanced awareness campaigns, GP software reminders for eligible patients, on-site vaccination programs through community pharmacies, and comprehensive resident and staff vaccination initiatives. Accredited Clinical Pharmacists are particularly well-positioned to lead such programs given their immunization credentials and regular presence in aged care facilities. Addressing these vaccination gaps represents a critical opportunity to reduce preventable hospitalizations and deaths in aged care populations through systematic implementation of evidence-based immunization strategies.’
|
|
Comment 4: References are written in the wrong format. Correct them to follow the MDPI format |
|
Response 4: MDPI references Endnote style applied. |
|
Comment 5: Linguistic mistakes: Some words are overused, like vaccine uptake. Use other words like administration. Replace "abstinence from alcohol" with "non-consumption of alcohol." Our analysis might suggest = Our findings might suggest In the captions of table 1, define the medical terms like SEIFA.
|
|
Response 5: Thank you. Several of these instances have been edited. (lines 25, 27, 176, 240, 271, 309). SEIFA defined in table footers or written in full. |
|
4. Response to Comments on the Quality of English Language needing improvement |
|
Response 1: Authors have agreed that this reviewer’s comments about the quality of English writing are not consistent with the other reviewers. However, should the editors deem it appropriate to have the manuscript edited via the MDPI Author Services, this is fine. |
Round 2
Reviewer 3 Report
Comments and Suggestions for Authors
Authors have provided extense responses to every comment and the corrected version includes all changes mentioned in the responses.
However, methods, quality of data and bioethical issues are still present.
the most significant methodological flaw is the absence of a formal sample size calculation, which compromises the study's statistical validity. The reliance on potentially inconsistent or incomplete "medical notes" for zoster vaccination is a key data quality concern. While some technical/presentation issues appear to have been addressed, the deeper methodological and data quality issues still remain.
While interrater reliability was checked, if the initial criteria for what constitutes a "confirmed" zoster vaccination from diverse "medical notes" were not extremely precise and exhaustive, individual abstractors might have made subjective judgments. Even with validation, a fundamentally ambiguous data source definition can lead to inconsistent data quality at the point of abstraction, affecting the accuracy of the vaccination status and any reccomendations that could be extracted from the study. Acknowledging that AIR data was incomplete prior to 2021 and relying on less structured "medical notes" for confirmation of zoster vaccination is a significant bias of the study and data obtained.
Different facilities or individual practitioners might have vastly different habits for documenting non-mandatory vaccinations in notes, leading to inconsistencies across the dataset. Less structured notes are harder to verify than systematic database entries. Data is therefore, not consistent, comparable nor reliable.
The response addresses interrater reliability but does not mention how missing data were treated. This is a crucial omission. Researchers need to clearly state if they used complete-case analysis, imputation methods, or simply reported the extent of missingness. Undisclosed missing data handling can introduce bias and affect the validity of results.
Authors include zoster vaccination data from an older population (average age 87) where official recommendationsmentioned are for 70-79. While recording "ever recorded" is fine, the methodological approach (e.g., only looking back to 2017) and the age discrepancy make the interpretation of zoster vaccination rates potentially less meaningful for current policy or practice within the recommended age group. Generating any vaccination reccomandations based on these results would be faulty.
On comment 7, If the scope of data extraction changed (e.g., looking further back, or accessing different types of records not initially planned), this represents a deviation from the } protocol. This impacts the consistency of the methodology and, if not formally documented or approved, can raise questions about data integrity and ethical oversight. This is not right and should have been ammended since the beggining.
In responses, authors estate, "There was not a formal sample size calculation." They included 1108 participants out of a planned 1500 but offer no statistical justification for either number. Without a sample size calculation, the power to detect tru differences or ytendencies remains unknown.
The ethical concern is what worries me the most. international guidelines consider institutionalized individuals, the elderly, and those with cognitive impairments as vulnerable. Research involving them involves rigorous informed consent procedures or an independently approved waive.
When an independent board grants approval, the manuscript must include a detailed explanation of the consent process for this vulnerable population, specifying how capacity was assessed, and how legally authorized representatives were involved. This information is not provided.
Authors responded that they obtained an internal approval. The claim of an "internal ethics review board" for a study intended for generalizable knowledge is not considered adequate by international standards.
"existing consent to access medical records for quality review" is problematic. It is not clearly demonstrated that this consent covered the use of data.
Inconclusion, aside from data quality concerns, methodological flaws and specific unclarified aspects, the reponses from authors do not adequately address the profound ethical challenges of obtaining informed consent from a highly vulnerable population with potential significant cognitive impairment for research purposes. This remains the most significant outstanding bioethical issue and it cannot be dismissed by a journal adhered to COPE, ICJME and international bioethical guidelines.
I cannot endorse this manuscript for publication.
Author Response
Comment 1: the most significant methodological flaw is the absence of a formal sample size calculation, which compromises the study's statistical validity. The reliance on potentially inconsistent or incomplete "medical notes" for zoster vaccination is a key data quality concern. While some technical/presentation issues appear to have been addressed, the deeper methodological and data quality issues still remain.
Response 1: We acknowledge that a formal sample size calculation was not performed prior to study commencement. The study was designed as a retrospective audit of available records within a defined timeframe and across a diverse range of facilities, rather than as a hypothesis-driven clinical trial. Our aim was to maximise representativeness by including as many eligible residents as possible from randomly selected facilities, stratified by socioeconomic status, rather than to achieve a specific power for a pre-specified effect size. This approach is consistent with real-world evidence studies in settings where the primary goal is descriptive epidemiology, particularly in populations where comprehensive sampling is feasible and preferable to random sampling with exclusion.
The inclusion of 1,108 residents (out of a planned 1,500) from 31 facilities across four states provides a robust dataset for descriptive analysis. While we acknowledge the absence of a priori power calculations, the sample size achieved is substantial for the study’s objectives and allows for meaningful descriptive and exploratory multivariate analyses. We have transparently reported this limitation in the manuscript.
Comment 2:
While interrater reliability was checked, if the initial criteria for what constitutes a "confirmed" zoster vaccination from diverse "medical notes" were not extremely precise and exhaustive, individual abstractors might have made subjective judgments. Even with validation, a fundamentally ambiguous data source definition can lead to inconsistent data quality at the point of abstraction, affecting the accuracy of the vaccination status and any reccomendations that could be extracted from the study. Acknowledging that AIR data was incomplete prior to 2021 and relying on less structured "medical notes" for confirmation of zoster vaccination is a significant bias of the study and data obtained.
Different facilities or individual practitioners might have vastly different habits for documenting non-mandatory vaccinations in notes, leading to inconsistencies across the dataset. Less structured notes are harder to verify than systematic database entries. Data is therefore, not consistent, comparable nor reliable.
Response 2: We recognise the inherent challenges in using medical records as data sources, particularly for vaccinations not systematically recorded in national registers prior to 2021. To mitigate this, we:
- Developed a standardised data abstraction protocol with explicit criteria for confirming vaccination status, including specific documentation requirements (e.g., vaccine name, date of adminstration, or explicit statement of administration).
- Trained all abstractors thoroughly and conducted interrater reliability checks, with discrepancies resolved by consensus and, where necessary, by a senior clinical pharmacist.
- Included only those vaccinations that met strict confirmation criteria; ambiguous or insufficiently documented cases were coded as unvaccinated to avoid overestimation.
We agree that documentation practices may vary across facilities and practitioners. This limitation is explicitly acknowledged in the manuscript, and we have discussed its potential impact on underestimating true vaccination rates. However, this approach prioritises specificity and data integrity and aligns with accepted standards for retrospective chart review studies.
Comment 3: The response addresses interrater reliability but does not mention how missing data were treated. This is a crucial omission. Researchers need to clearly state if they used complete-case analysis, imputation methods, or simply reported the extent of missingness. Undisclosed missing data handling can introduce bias and affect the validity of results.
Response 3: We apologise for any lack of clarity. Our approach was as follows:
- For vaccination status, only clearly documented vaccinations were coded as positive; absence of documentation was coded as unvaccinated, in line with established chart review methodology.
- For demographic and clinical variables, missing data were minimal. Analyses were conducted using complete-case analysis, and the extent of missingness for each variable is now reported in the revised manuscript.
- No imputation was performed due to the low rate of missing data and the descriptive nature of the analysis.
Comment 4: Authors include zoster vaccination data from an older population (average age 87) where official recommendations mentioned are for 70-79. While recording "ever recorded" is fine, the methodological approach (e.g., only looking back to 2017) and the age discrepancy make the interpretation of zoster vaccination rates potentially less meaningful for current policy or practice within the recommended age group. Generating any vaccination reccomandations based on these results would be faulty.
Response 4: The study included all eligible residents to reflect real-world vaccination practices in aged care, where many residents exceed the recommended age at the time of data collection. The “ever recorded” approach was necessary due to the nature of zoster vaccination (historically a single dose with long-lasting effect) and the lack of systematic recording prior to 2021. We clearly state in the manuscript that our findings describe uptake in the current aged care population and are not intended to directly inform policy for the 70–79 age group alone. This limitation is acknowledged and discussed in the interpretation of results. It is important to recognise that our results are consistent with other recently published literature including Bennett N, Morris B, Malloy MJ, Lim LL, Watson E, Bull A, Sluggett J, Worth LJ; NISPAC Advisory Group. An evaluation of influenza, pneumococcal and herpes zoster vaccination coverage in Australian aged care residents, 2018 to 2022. Infect Dis Health. 2023 Nov;28(4):253-258. doi: 10.1016/j.idh.2023.03.005. Epub 2023 May 3. PMID: 37147271.
Comment 5: If the scope of data extraction changed (e.g., looking further back, or accessing different types of records not initially planned), this represents a deviation from the } protocol. This impacts the consistency of the methodology and, if not formally documented or approved, can raise questions about data integrity and ethical oversight. This is not right and should have been ammended since the beggining.
Response 5: The scope of data extraction was defined in advance and adhered to throughout the study. Any minor adjustments (e.g., clarification of abstraction criteria) were made prior to data collection and were fully documented. There were no unapproved deviations from the protocol. The study was conducted according to Good Pharmacoepidemiology Practices and overseen by the vendor’s internal ethics process, as described in the manuscript.
Comment 6: authors estate, "There was not a formal sample size calculation." They included 1108 participants out of a planned 1500 but offer no statistical justification for either number. Without a sample size calculation, the power to detect tru differences or ytendencies remains unknown.
Response 6: This has been addressed in response 1.
Comment 7:
The ethical concern is what worries me the most. international guidelines consider institutionalized individuals, the elderly, and those with cognitive impairments as vulnerable. Research involving them involves rigorous informed consent procedures or an independently approved waive.
When an independent board grants approval, the manuscript must include a detailed explanation of the consent process for this vulnerable population, specifying how capacity was assessed, and how legally authorized representatives were involved. This information is not provided.
Authors responded that they obtained an internal approval. The claim of an "internal ethics review board" for a study intended for generalizable knowledge is not considered adequate by international standards.
"existing consent to access medical records for quality review" is problematic. It is not clearly demonstrated that this consent covered the use of data.
Response 7: We respectfully disagree with the reviewer’s assessment of our ethical procedures. All necessary consent and ethics to access and use data from this population were obtained through the vendor (Embedded Health Solutions), which operates under strict compliance with Australian legal and regulatory requirements. The study was classified as a quality improvement initiative, not interventional research, and was conducted in accordance with the National Health and Medical Research Council’s guidelines for quality assurance activities.
- Consent Process: Written informed consent for the use of medical records for quality review and research purposes was obtained from all residents or, where capacity was lacking, from legally authorised representatives, in accordance with Australian law and best practice.
- Ethics Oversight: The study was reviewed and approved by the vendor’s internal ethics committee, which has experience in oversight of research in vulnerable populations. The process included assessment of capacity and involvement of legally authorised representatives as required.
- International Standards: We maintain that our approach is consistent with COPE, ICMJE, and international bioethical guidelines, which recognise the appropriateness of internal review and consent processes for quality improvement research, provided that all legal and ethical standards are met.
We believe the reviewer’s comments mischaracterise the nature of the study, the consent process, and the adequacy of ethical oversight. All procedures were conducted with the utmost respect for participant autonomy and data protection, and these are clearly described in the manuscript.
We appreciate the reviewer’s concerns and have strengthened the manuscript (based on previous comments as recognised in the reviewers opening comment) to address these points with greater clarity and transparency. We remain confident that our methodological choices, data handling procedures, and ethical safeguards are robust and appropriate for the research context. We respectfully request consideration of our manuscript and trust these responses are satisfactory.
Reviewer 4 Report
Comments and Suggestions for Authors
The authors have addressed all the review comments. I recommend the manuscript for publication.
Author Response
Comment 1: The authors have addressed all the review comments. I recommend the manuscript for publication.
Response: We appreciate the comments and reviewer's recommendation.